# Selection for and Analysis of UV-Resistant Cryptophlebia Leucotreta Granulovirus-SA as a Biopesticide for *Thaumatotibia leucotreta*

**DOI:** 10.3390/v14010028

**Published:** 2021-12-24

**Authors:** Patrick Mwanza, Michael Jukes, Gill Dealtry, Michael Lee, Sean Moore

**Affiliations:** 1Department of Physiology, Nelson Mandela University, Gqeberha 6031, South Africa; patrick.mwanza@mandela.ac.za; 2Centre for Biological Control, Department of Zoology and Entomology, Rhodes University, Makhanda 6139, South Africa; m.jukes@ru.ac.za (M.J.); seanmoore@cri.co.za (S.M.); 3Department of Biochemistry and Microbiology, Rhodes University, Makhanda 6140, South Africa; 4Centre for HRTEM, Nelson Mandela University, Gqeberha 6001, South Africa; michael.lee@mandela.ac.za; 5Citrus Research International, Gqeberha 6065, South Africa

**Keywords:** baculovirus, climate chamber, concentration-response bioassay, single nucleotide polymorphism

## Abstract

Cryptophlebia leucotreta granulovirus-SA (CrleGV-SA) is used as a commercial biopesticide for the false codling moth, *Thaumatotibia leucotreta*, in citrus and other crops. The virus is sensitive to UV irradiation from sunlight, which reduces its efficacy as a biopesticide in the field. We selected a UV-resistant CrleGV-SA isolate, with more than a thousand-fold improved virulence compared to the wild-type isolate, measured by comparing LC_50_ values. CrleGV-SA purified from infected *T. leucotreta* larvae was exposed to UV irradiation under controlled laboratory conditions in a climate chamber mimicking field conditions. Five cycles of UV exposure, followed by propagating the virus that retained infectivity in vivo with re-exposure to UV, were conducted to isolate and select for UV-resistant virus. Serial dilution bioassays were conducted against neonates after each UV exposure cycle. The concentration-responses of the infectious UV-exposed virus populations were compared by probit analysis with those from previous cycles and from the original CrleGV-SA virus population. NGS sequences of CrleGV-SA samples from UV exposure cycle 1 and cycle 5 were compared with the GenBank CrleGV-SA sequence. Changes in the genomes of infective virus from cycles 1 and 5 generated SNPs thought to be responsible for establishing UV tolerance. Additional SNPs, detected only in the cycle 5 sequence, may enhance UV tolerance and improve the virulence of the UV-tolerant population.

## 1. Introduction

Biological control of agricultural pests has gained popularity globally over recent years due to the pressures to reduce the use of agrochemicals in the environment and their residues in foodstuffs, and the need for alternative controls to combat pest resistance to chemical pesticides [1]. Biological control involves the conservation, introduction, or augmentation of natural enemies or pathogens, such as bacteria, fungi, and viruses, within an environment to control a species that has attained pest status [1,2]. These biopesticides are generally specific to their target organisms, and are therefore safe for beneficial insects, as well as for human consumption [3]. Biological control methods form part of integrated pest management (IPM) programs, which incorporate cultural, physical, chemical, and biological methods to control pests [4].

Baculoviruses are registered biopesticides for many insect pests [3,5,6], including the control of several tree fruit Lepidoptera. Individual baculoviruses are highly specific to their host, with most baculovirus species being pathogenic to a single insect species or closely related species, and hence they are safe for vertebrates and other beneficial organisms [7]. *Thaumatotibia leucotreta* (Meyrick) (Lepidoptera: Tortricidae) [8], known as the false codling moth, is a pest for citrus and some other crops and is controlled by Cryptophlebia leucotreta granulovirus (CrleGV) [9,10]. CrleGV biopesticides are registered in South Africa for use on citrus, avocadoes, macadamias, grapes, and other crops by two commercial producers, River Bioscience (SA) and Andermatt (Switzerland), as part of an IPM program for *T. leucotreta* in citrus. *Thaumatotibia leucotreta* has major economic significance due to its phytosanitary status. Detection of a single larva in fruit marked for export could result in the consignment being rejected [11,12,13]. Therefore, management of this pest pre-and post-harvest is critical to citrus exports from South Africa.

A limitation on the use of all baculovirus biopesticides is their sensitivity to ultraviolet (UV) irradiation, leading to their rapid degradation in direct sunlight and loss of activity within hours to a few days of UV exposure [14,15]. Although the occlusion body (OB) protects the virion from many environmental factors, it does not offer protection against UV [14,15]. UV irradiation damages baculoviruses by cross-linking adjacent pyrimidine residues in the DNA, resulting in inhibition of DNA replication, mutation, and blocking transcription factors [16,17]. Direct DNA damage also causes deletions, strand breakage, and labile sites in the DNA [18]. In addition, UV-generated reactive oxygen species such as peroxides, single oxygen, or hydroxyl radicals inactivate the OBs [19,20].

The total amount of UV incident in a field varies, depending on the geography and the season [21]. Under field conditions, the half-life of baculoviruses varies from 10 h to 10 days, with the average half-life being around 24 h in the absence of any form of UV protection [22,23]. The need for UV protection is determined by the crop plant architecture and by where the pest feeds [24]. We have shown that degradation of CrleGV in the field is more rapid on the northern (sun-facing) side of the crop plant than on the southern side [11,15]. UV inactivates baculoviruses faster in wet suspension [25]; therefore, most spraying in the field is done in the evening to reduce the impact of UV irradiation [26]. Nevertheless, as a result of the effects of UV radiation, South African farmers must respray with CrleGV formulation up to three times per season between November and March, and sometimes even more frequently.

Therefore, we have undertaken to select a UV-resistant strain of CrleGV from the CrleGV-SA isolate used in current biopesticides. Other studies have demonstrated differences in the UV tolerance of baculovirus species, and of different isolates of the same baculovirus from different geographical regions [27,28]. Akhanaev et al. [28] compared the UV tolerance of two LdMNPV strains from Western Siberia (LdMNPV-27/0) and North America (LdMNPV-45/0) by measuring the relative rate of inactivation and virus half-life. The North American strain, previously shown to be more virulent towards *Lymantria dispar* larvae, was more sensitive to UV following 15 min of sunlight exposure and lost its potency faster than the Siberian strain. Witt and Stairs [29] showed that within a population of *Galleria mellonella* NPV (GmNPV) active against *Galleria mellonella*, a sub-population was susceptible to low doses of UV, while another sub-population was susceptible only to high UV dosage, translating to almost a thousand-fold difference in UV susceptibility. They postulated that this heterogeneity in UV response could be the result of genetic variability and that it would be possible to select strains of virus that are UV-tolerant.

Therefore, we have undertaken to select a UV-resistant strain of CrleGV from the CrleGV-SA isolate used in current biopesticides. Other studies have demonstrated differences in the UV tolerance of baculovirus species, and of different isolates of the same baculovirus from different geographical regions [27,28]. Akhanaev et al. [28] compared the UV tolerance of two LdMNPV strains from Western Siberia (LdMNPV-27/0) and North America (LdMNPV-45/0) by measuring the relative rate of inactivation and virus half-life. The North American strain, previously shown to be more virulent towards *Lymantria dispar* larvae, was more sensitive to UV following 15 min of sunlight exposure and lost its potency faster than the Siberian strain. Witt and Stairs [29] showed that within a population of *Galleria mellonella* NPV (GmNPV) active against *Galleria mellonella*, a sub-population was susceptible to low doses of UV, while another sub-population was susceptible only to high UV dosage, translating to almost a thousand-fold difference in UV susceptibility. They postulated that this heterogeneity in UV response could be the result of genetic variability and that it would be possible to select strains of virus that are UV-tolerant.

We have shown, in laboratory experiments aimed at determining the reapplication frequency of CrleGV-SA formulations, that residual activity remains in bioassays conducted with CrleGV-SA exposed to UV under controlled conditions, even after 7 days’ exposure [15]. In field studies following spraying of CrleGV-SA biopesticide in citrus orchards, Mwanza [15] recorded that 21 days after spraying there was a significantly lower virus LD_50_ in samples from the southern (shady) side of citrus trees than in samples from the northern (sunny) side of the trees. One week later (i.e., 28 days after spraying), virulent virus was detected in samples from the northern side of the trees, but the activity was too low to quantify. In comparison, virus samples from the southern side of trees gave a clear bioassay concentration response. This persistence is partly due to the architecture of citrus trees, which provide more shading than most crops on which baculovirus biopesticides are applied [9]. It is also possible that this was due to inherent UV resistance in some of the viruses in the biopesticide, even when subjected to direct UV radiation on the northern side. This indicates the potential for selection of UV-tolerant CrleGV strains that would enable development of a biopesticide that persists longer in the field.

Several isolates of CrleGV have been identified. A natural isolate of CrleGV was first identified from the Ivory Coast (CrleGV-IC) [30]; another isolate was from Cape Verde (-CV) [31]; and at least seven genetically distinct CrleGV-SA isolates were from South Africa (-SA), one of which is used in the formulation of the commercial biopesticide produced in South Africa [32]. This indicates a level of variation in the CrleGV virus genomes. It is conceivable that these natural variants of CrleGV may also have differing UV sensitivity and that it would be possible to select a UV-resistant isolate from CrleGV-SA by repeated exposure to UV under laboratory conditions. This approach has been successfully followed with other baculoviruses. An early study was conducted by Brassel and Benz [33], who reported a six-step selection process that yielded a strain of CpGV with a 5.6-fold increase in UV tolerance over the original strain. Each cycle involved a UV exposure step followed by in vivo propagation of the virus that retained infectivity. Other researchers have shown a 2.5-fold increase in virus persistence after 6 cycles of UV exposure and propagation of the gypsy moth NPV, Lymantria dispar multiple nucleopolyhedrovirus (LdMNPV) [34]; a half-life increase for Phthorimaea operculella (Zeller) granulovirus (PhopGV) from 2.6 to 24 min after four UV exposure cycles at a total irradiation of 1100 W/m^2^ [35]; and increased UV tolerance and retention of virulence from the third UV exposure cycle upwards for Helicoverpa armigera NPV (HearNPV coimbatore isolate CBE 1) [36]. A new variant of Adoxophyes orana granulovirus (AdorGV), designated AdorGV-M, isolated from *Adoxophyes* spp. larvae in the field, was shown to be as pathogenic as an English isolate AdorGV-E, and to have a five-fold longer half-life than AdorGV-E after UV irradiation [37]. The AdorGV-M isolate had significantly larger cuboidal OBs, as opposed to the usual ovo-cylindrical shape associated with most granuloviruses. The larger OBs were thought to contribute to this difference in UV tolerance, as the larger OBs provided a thicker layer of crystalline protein matrix than those of AdorGV-E [37].

The main objective of this study was to select a UV-resistant or tolerant isolate of CrleGV-SA that would provide longer lasting virulence of a biopesticide in the field.

## 2. Materials and Methods

### 2.1. Virus Purification

Fifth instar *T. leucotreta* larvae were inoculated with the known isolate of CrleGV-SA used in a formulation of the biopesticide, at the appropriate LC_90_ concentration [38]. Larvae displaying infection symptoms were isolated and stored at −20 °C until virus was required for experimentation. Viruses were isolated and purified from the larval cadavers following the protocol of Hunter-Fujita et al. [39] modified by Moore [11]. The virus pellet obtained from homogenised larvae was separated by centrifugation on a 30–80% glycerol gradient in an Optima Ultracentrifuge Beckman L70 rotor (Beckman Coulter, Brea, CA, USA) at 40,572 g for 15 min to obtain a pure virus band that was washed and resuspended in 8 mL double distilled water at 4 °C.

### 2.2. Virus Enumeration

Purified virus diluted in 0.1% sodium dodecyl sulphate (SDS) in double distilled water was enumerated in a 0.02 mm deep Helber bacterial counting chamber (Hawksley Medical and Laboratory Equipment, Lancing, UK) under dark field microscopy, as described by Hunter-Fujita et al. [39] using an Olympus BX 51 TF microscope (Olympus, Tokyo, Japan). All preparations were enumerated in triplicate and the mean counts were used to determine virus OB concentration, using the formula OB mL^−1^ = (dilution × mean OB count)/(80 × (5 × 10^−8^)), where 80 is the number of small squares counted and 5 × 10^−8^ is the volume in millilitres of the virus suspension within these squares.

### 2.3. Selection of UV-Resistant CrleGV-SA

*Thaumatotibia leucotreta* egg sheets and fifth instars were provided by River Bioscience, Gqeberha, SA from their Addo (Eastern Cape, South Africa) insect rearing facility. The eggs were incubated at 28 °C in glass jars for 24–48 h until hatched. First instar larvae were used in surface concentration bioassays within 24 h of hatching. Fifth instar larvae were used for the virus propagation steps. UV exposure was carried out in a Q-Sun Xe-3 HC test chamber (Q-lab, Westlake, OH, USA), fitted with three 100 W xenon arc lamps and a Daylight Q optical filter, which mimics UV conditions of normal sunlight, with irradiance set at 300 Wm^−2^. Temperature and relative humidity were maintained at 30 °C and 42%. These conditions were based on averages collected over one summer period in the Sundays River Valley, Eastern Cape Province, an important citrus growing area (Linta Greef, Sundays River Citrus Company, Kirkwood, SA, personal communication). Purified aliquots of 3 mL CrleGV-SA at a concentration of 1 × 10^10^ OB mL^−1^ were dried overnight in petri dishes within a laminar flow hood and then placed in the UV test chamber for 1, 3, 8, 24, and 72 h. The UV-exposed virus samples were then resuspended in 3 mL double distilled water, quantified, and stored at 4 °C until needed for propagation for the next cycle, or for bioassay.

After each exposure to UV light, the resultant virus sample was fed to fifth instars and the infective virus was amplified within the larvae. Individual larvae were reared on artificial diet with a surface inoculation of 50 µL UV-exposed CrleGV-SA in 25 well bioassay plates incubated at 30 °C. Infected larvae, dead or almost dead, were collected over a 14-day period and stored at −20 °C. Virus was extracted from the larvae and purified as described in Section 2.1 and stored at 4 °C until the next exposure cycle, or for analysis. The process of UV exposure and subsequent propagation of infective virus in fifth instars constituted one exposure cycle. A total of five exposure cycles were carried out.

### 2.4. Concentration Bioassay

Concentration response bioassays were conducted on first instars in 25 well bioassay plates, according to a standard protocol [38]. UV-exposed virus samples for each exposure time point were adjusted to 1 × 10^9^ OB mL^−1^ and serially diluted five-fold to give five concentrations for each time point, ranging from 3 × 10^7^ OB mL^−1^ to 3 × 10^3^ OB mL^−1^. Aliquots of 50 μL per well of a single virus concentration were spread on the surface of the diet and dried under a laminar flow hood for 30 min; a single neonate larva was placed on the surface of the diet in each well and incubated at 28 °C for 7 days, after which larval mortality was recorded. Each bioassay was carried out in triplicate using 25 larvae per concentration for each replicate. A negative control plate with sterile double distilled water and plates with CrleGV-SA not exposed to UV were included. All bioassays were carried out in triplicate. The mean mortality data obtained were subjected to probit analysis using PROBAN, a statistical software program used for analysis of bioassay data [40]. This software takes into consideration the mortality of the treated larvae and corrects for the mortality of control larvae, based on the Abbot formula [41], giving a concentration response curve from which the LC_50_ values were determined at each exposure time. PROBAN transformed the doses to log_10_ and the percentage mortality response to empirical probits. Regression lines comparing responses at a given time point across the five cycles were determined and the slopes of the lines were compared. Significant differences at *p* ≤ 0.05 were determined. If lines were found to be parallel, relative potency comparisons were carried out. For each comparison at each time point, the cycle 1 sample was chosen as the reference (r) and compared with another sample (*t*) from a different cycle at the same time point. If *t* was less than 1 (*t* < 1), the test sample was more potent than the reference sample; if the value of t equalled 1 (*t* = 1), there was no difference in potency between the two samples; a value of t greater than 1 (*t* > 1) indicated that the test sample was less potent than the reference sample [40,42,43,44,45]. Bartlett’s test was used to compare the homogeneity of variances in the lines at *p* ≤ 0.01. The Chi-square test and the Bonferroni method were used to determine that the lines were parallel, and whether the elevations were comparable in each cycle.

### 2.5. DNA Extraction

CrleGV-SA samples obtained from 1 cycle of UV exposure for 72 h, and 5 cycles of UV exposure for 72 h, were amplified in fifth instars. OBs were extracted, purified, and diluted to a concentration of 1 × 10^8^ OB mL^−1^ and genomic DNA was extracted using a CTAB extraction method described by Singh et al. [46], modified by Goble [47]. The DNA pellet was air-dried and resuspended in 50 μL RNase-free, DNase-free ultrapure water and stored at −20 °C.

### 2.6. DNA Sequencing

Approximately 200 ng genomic DNA extracted from CrleGV-SA samples from UV exposure cycle 1 (CrleGV-SA C1) and cycle 5 (CrleGV-SA C5) were sequenced by Inqaba Biotec, SA, using next generation DNA sequencing (NGS) on the MiSeq desktop sequencer (Illumina Inc., San Diego, CA, USA). The reads for each sample were paired, error-corrected, and normalized using the BBNorm plugin in Geneious R11 (Biomatters Ltd., Auckland, New Zealand), with a read depth of 1000 and 2000 set for C1 and C5, respectively. Reads were subsequently assembled into contigs via de novo assembly in Geneious R11 at medium-low sensitivity. The CrleGV-SA genome (GenBank Accession number MF974563 [48]) was used as the reference sequence, to which contigs were dissolved and reassembled forming a single contig for each of the two samples. Medium sensitivity was used for assembly of the reads and single consensus sequences were generated for CrleGV-SA C1 and CrleGV-SA C5. Pairwise multiple alignments were performed on the consensus sequences, and thereafter predicted open reading frames (ORFs) were mapped against the reference CrleGV-SA published sequence. The Find SNPs/Variants tool was used to search for single nucleotide polymorphisms (SNPs) in both CrleGV-SA C1 and CrleGV-SA C5.

## 3. Results

### 3.1. Surface Concentration Bioassays

In bioassays, the mortality of larvae was related to the concentration of the control (0 h UV-exposure) and all five UV-exposure time points for each cycle, and concentration response relationships were determined. Control mortality of samples for each cycle ranged from 0% to 13%. The dead larvae in these control bioassays did not exhibit symptoms of viral infection and were not considered to be covert infections. The regression lines fitted to the corrected data for all replicates were compared for each cycle and the residual variances of the lines were determined by Bartlett’s test. The variances were determined to be homogeneous (Χ^2^ = 0.763, DF = 5, *p* = 0.01, cycle 1; Χ^2^ = 0.214, *p* = 0.01, cycle 2; Χ^2^ = 1.382, *p* = 0.01, cycle 3; Χ^2^ = 0.237, *p* = 0.01, cycle 4; Χ^2^ = 0.237, *p* = 0.01, cycle 5), and thus comparisons of slopes and elevations could be carried out. The lines were determined to be parallel by the Chi-square test, and their elevations were shown to be comparable (Χ^2^ = 8.208, DF = 5, *p* = 0.05, cycle 1; Χ^2^ = 4.642, *p* = 0.05, cycle 2; Χ^2^ = 3.773, *p* = 0.05, cycle 3; Χ^2^ = 0.602, *p* = 0.05, cycle 4; Χ^2^ = 0.602, *p* = 0.05, cycle 5). The Bonferroni method used to compare the elevations of the lines determined that the elevations differed significantly from each other in each cycle (F_5, 23_ = 2.64, *p* = 0.05, cycle1; F_5, 23_ = 2.64, *p* = 0.05, cycle 2; F_5, 23_ = 2.64, *p* = 0.05, cycle3; F_5, 23_ = 2.64, *p* = 0.05, cycle 4; F_5, 23_ = 2.64, *p* = 0.05, cycle 5). The regression lines were parallel; therefore, relative potency comparisons of the LC_50_ values were determined.

Following one cycle of UV exposure, the LC_50_ values increased from 2.29 × 10^4^ OB mL^−1^ for the non-irradiated control to 2.11 × 10^9^ OB mL^−1^ after 72 h of UV exposure (Table 1). After two UV exposure cycles, LC_50_ values also increased from 2.57 × 10^4^ OB mL^−1^ for the non-irradiated control to 1.59 × 10^9^ OB mL^−1^ after 72 h of UV exposure (Table 1). After three cycles, the LC_50_ values at each time point decreased in comparison to the corresponding time points in exposure cycle 2. The non-irradiated control LC_50_ value was 2.06 × 10^4^ OB mL^−1^ (comparable to the previous cycles); after 3 h UV-exposure, this increased to 1.18 × 10^6^ OB mL^−1^; after 8 h UV-exposure, the LC_50_ was 4.26 × 10^5^ OB mL^−1^; after 24 h exposure, it was 1.15 × 10^7^ OB mL^−1;^ and finally, it was 8.18 × 10^6^ OB mL^−1^ after 72 h UV exposure (Table 1). Relative potency comparisons for the 24 h UV-exposure samples showed the cycle 3 virus sample was more potent than the cycle 2 (*t* = 0.699) and cycle 1 samples (*t* = 0.027) (Table 2). Similarly, the 72 h UV-exposure sample from cycle 3 was more potent than the corresponding cycle 2 (*t* = 1.262) and cycle 1(*t* = 0.004) samples (Table 2). Following the fourth cycle of UV exposure, LC_50_ values increased from 2.08 × 10^4^ OB mL^−1^ for the non-irradiated control to 1.47 × 10^6^ OBs/mL for the 3 h UV-exposure sample; 5.36 × 10^5^ OB mL^−1^ for 8 h UV exposure; 1.22 × 10^7^ OB mL^−1^ after 24 h UV exposure; and finally, 6.12 × 10^6^ OB mL^−1^ after 72 h UV exposure (Table 1). Relative potency comparisons showed the cycle 4 sample to be more potent than the cycle 3 virus sample (*t* = 0.002) after 24 h exposure to UV (Table 2). This increased relative potency was also evident in the samples exposed for 72 h (*t* = 0.004) (Table 2). LC_50_ values for virus after cycle 5 increased from the non-irradiated control LC_50_ of 2.87 × 10^4^ OB mL^−1^) to the 8 h UV-exposure sample 6.38 × 10^6^ OB mL^−1^. After 24 h UV exposure, the LC_50_ dropped to 2.16 × 10^5^ OB mL^−1^ but increased to 1.73 × 10^6^ OB mL^−1^ after 72 h UV exposure (Table 1). The 72 h sample from cycle 5 was selected for further molecular and structural analysis, as it showed the greatest change in LC_50_ values after UV re-exposure.

Comparison of cycle 1 virus with cycle 5 virus following probit analysis of the concentration response in bioassays of virus exposed to UV for 24 and 72 h, indicates a 1338-fold reduction in LC_50_ at 24 h UV exposure after five cycles of selection, compared to just one cycle, and a 1227-fold reduction in C5 virus LC_50_ at 72 h exposure.

### 3.2. NGS Sequencing of Cycle 1 and Cycle 5 Samples Exposed to UV for 72 h

Sequencing of CrleGV-SA C1 genome generated 416,314 paired reads, with error correction and normalization reducing this figure to 279,938. From this, 278,399 reads were used to produce 470 contigs. The largest contig was 115,445 bases long and was assembled from 272,009 sequences. The CrleGV-SA cycle 1 genome was assembled into a contiguous sequence with a length of 111,334 bp, a GC content of 32.6%, and 99.9% identity to the published CrleGV-SA genome [48]. A mean read depth of 549.2 (±53.7) was achieved with Q40, Q30, and Q20 scores of 98.6%, 99.1%, and 99.6%, respectively.

The CrleGV-SA C5 generated 26,985,586 paired reads, which reduced to 1,035,796 following error correction and normalization. Of these, 1,030,337 reads produced 3901 contigs. The largest contig was 56,825 bases long and was assembled from 492,215 sequences. The CrleGV-SA C5 genome was assembled into a contiguous sequence with a length of 111,334 bp and a mean read depth of 1244.8 (±127.6) from reads with Q40, Q30, and Q20 scores of 98.9%, 99.1%, and 99.7%, respectively. The resultant nucleotide alignment had a GC content of 32.6% and 99.99% identity to the published CrleGV-SA genome. No polymorphism was observed in either the C1 or the C5 genome. Seven non-synonymous SNPs were detected after mapping of the CrleGV-SA C1 sequence and the CrleGV-SA C5 sequence to the published unexposed CrleGV-SA genome sequence (Table 3). The first SNP, at position 434, was a transition, where guanine was changed to adenine in the granulin gene. This would result in an amino acid change from the sulfur-rich cysteine to the acidic tryptophan. A second SNP, at position 36,843, involved a transversion from adenine to thymine in the metalloproteinase coding sequence (CDS). This SNP resulted in a change of amino acid from the aromatic phenylalanine to the aliphatic isoleucine. At nucleotide position 38,194, an SNP transition from thymine to cytosine resulted in the change of amino acid from isoleucine to the hydroxylic threonine. At position 45,853, an SNP transition from cytosine to thymine resulted in the amino acid changing from the aliphatic valine to the sulfur containing methionine. An SNP transition at position 79,840 resulted in the change of amino acid from valine to leucine. Another amino acid change from the acidic glutamic acid to the basic lysine resulted from an SNP transition from guanine to adenine at position 94,086. The last SNP was detected at position 104,574, where thymine was replaced by cytosine, resulting in the change of amino acid from methionine to threonine and, consequently, the loss of a start codon in a hypothetical CDS.

Mapping of the CrleGV-SA C5 sequence to the published sequence of the unexposed CrleGV-SA genome detected a total of 14 non-synonymous SNPs, of which 7 were the same as those identified in the cycle 1 sequence (Table 3) and an additional 7 were unique to the cycle 5 sequence (Table 4). The first of the unique SNPs was at position 13,168, where a cytosine was replaced by thymine, which led to the amino acid change from alanine to valine. At position 59,709–59,710, two thymine residues were replaced by two cytosine residues, leading to the amino acid change from isoleucine to valine. At 59,734, adenine was replaced by thymine, leading to the change in amino acid from aspartic acid to glutamic acid. At 59,752, an SNP transversion replaced an adenine residue by a cytosine residue, resulting in a change in amino acid from the basic histidine to the amidic glutamine in the DNA binding protein (ORF-72) gene. This gene partially overlaps with the hypothetical protein (ORF-73) sequence at position 59,752, hence both are indicated in Table 4. At 59,779, a thymine was replaced by a cytosine, which led to the amino acid change from the hydroxylic serine to the aliphatic serine, also in ORF-73. At 78,522, a guanine was replaced by an adenine and consequently a serine amino acid was replaced by a phenylalanine.

## 4. Discussion

Management of the pest *T. leucotreta* pre- and post-harvest is critical to citrus exports from South Africa, with the baculovirus CrleGV-SA used as part of an IPM program. The major shortcoming of this and other baculoviruses is probably UV sensitivity, resulting in rapid breakdown when directly exposed to sunlight [49]. Despite this, the virus can be surprisingly persistent after application in the field, with efficacy being recorded for up to 17 weeks after application [26]. Moore and co-workers [9,26] surmised that there are four reasons for the protracted CrleGV persistence recorded on citrus. First, a citrus tree provides substantial shading and therefore protection of virus against UV inactivation, more than probably any other crops on which viruses have been tested for pest control. Second, it has been observed that during most of the growing season, the majority of *T. leucotreta* larvae penetrate a Navel orange through its navel end. It is precisely here that CrleGV could be well-protected against sunlight and possibly even rainfall. 

The remaining two reasons for this recorded persistence are secondary. *T. leucotreta* takes a long time to recolonize an area, even after the efficacy of a spray might have expired. This slow migration was confirmed by Timm et al. [50] and Stotter et al. [51]. Finally, as CrleGV would have little, if any, detrimental impact on the highly effective and naturally occurring egg parasitoid, *Trichogrammatoidea cryptophlebiae* [52,53], this biocontrol agent could aid in maintaining control of *T. leucotreta* when virus is no longer effective. However, such protracted persistence cannot be accepted as the norm. Where trees are young or less dense and on cultivars other than Navel oranges, the virus will be far more exposed to UV irradiation. Rapid degradation and loss of activity was recorded in laboratory trials, and in field trials there was a 36-fold increase in LD_50_ after 7 days’ exposure on the sunny northern side of trees [15]. Some residual virus activity was recorded in both the laboratory and field experiments, which may be due to inherent UV resistance in the virus population [15].

Several naturally occurring isolates of CrleGV have been identified in the Ivory Coast (CrleGV-IC) [30], Cape Verde (CrleGV-CV) [31], and South Africa (CrleGV-SA) [32], indicating a level of variation in the CrleGV virus genomes and possible differing UV sensitivity. This suggested the potential for selection of a UV-resistant or UV-tolerant CrleGV strain, by repeated exposure and selection of the CrleGV-SA isolate currently used in commercial biopesticides to UV under laboratory conditions. Such a process would enable development of a biopesticide that persists longer in the field. Repeated exposure to UV, followed by selection of active virus, has been successfully undertaken with other baculoviruses to increase UV resistance [33,34,35,36,37].

In the present study, the South African isolate of (CrleGV-SA) was exposed to UV irradiation for 5 exposure cycles in a Q-Sun Xe-3 HC test chamber (Q-lab, Westlake, OH, USA) with parameters set to mimic a typical summer day in the Sundays River Valley, Eastern Cape Province, South Africa. Between exposures, virus that retained infectivity was multiplied in *T. leucotreta* fifth instars. Surface concentration bioassays were conducted to determine the LC_50_ of the virus after each exposure cycle. Virus samples exposed to UV in cycle 5 had lower LC_50_ values compared to virus samples from the early cycles. With each re-exposure cycle, the LC_50_ values moved closer to the value of the unexposed control. Thus, a UV-resistant or UV-tolerant isolate was selected after 5 cycles of UV exposure. Whether this isolate is resistant to UV or tolerant of UV cannot yet be determined, and it is not clear how stable this tolerance or resistance to UV is. Confirmation of maintenance of genetic stability of the isolate after further in vivo replication would indicate this tolerance or resistance. However, the virus remains virulent after longer exposure to UV than the original isolate, and therefore is of value as an improved biopesticide. The LT values for the UV-exposed virus from cycle 1 and cycle 5 were not determined in these bioassays; however, this will be addressed in future experiments, as the time to death of the larvae is an important factor in biocontrol applications.

To characterize the changes in the CrleGV-SA genome occurring during the selection process, DNA samples from viruses purified after exposure cycles 1 and 5 were sequenced by NGS. The resultant sequence data were compared with the published CrleGV-SA full genome sequence, as determined by van der Merwe et al. [48] and the CrleGV-CV3 isolate sequenced by Lange and Jehle [54]. Analysis of the CrleGV-SA UV-resistant isolates identified seven non-synonymous SNPs in cycle 1, which may help establish UV tolerance. A further seven SNPs were identified in cycle 5 samples. We propose that these cycle 5 SNPs further establish and maintain UV tolerance and may be associated with virulence, as evidenced by the bioassay data in which the LC_50_ is reduced in cycle 5 isolates. The SNPs were largely substitutions and did not consist of other variations, such as deletions or insertions. These SNPs occurred in regions of known proteins, as well as in hypothetical protein regions. Some of the SNPs were found in genes that regulate or are involved in the infection cycle, such as the *pif-2* and the metalloproteinase genes. This could explain the reduction in LC_50_ of virus selected after the fifth cycle of UV exposure. Another SNP was found in the granulin gene that encodes the major protein forming the OB. This could improve the stability of the protein, or potentially confer UV protective capacity to the OB by influencing the crystalline structure of the OB. Additionally, the SNPs could lie in regulatory sequences or affect codon usage, and may affect mRNA structure, folding, or stability. However, van der Merwe et al. [48] demonstrated that the CrleGV-SA genome has remained stable in the past 15 years, and therefore we conclude that the differences in UV tolerance demonstrated in the present study are likely to have been generated by the repeated UV exposure and re-exposure of the virus. It is possible that some low frequency variants of the virus may have re-emerged under harsh UV selection pressure; for example, the pif-1 SNP has been previously detected in wild-type populations [48]. The virus used for sequencing was not genetically clonal, but no polymorphism was observed in the genomes of cycle 1 or cycle 5 virus. Several rounds of passage of the cycle 5 UV-tolerant virus in the larval host, without further UV treatment, would confirm that it is a single isolate population. Re-testing for virulence after further UV exposure would then confirm that the passaged virus was still tolerant to UV.

TEM data from CrleGV-SA [55] showed that UV damages the virion, as well as the crystalline structure, of the OB. Comparison of cycle 1 and cycle 5 UV-exposed OBs revealed that the cycle 5 OBs were significantly larger than the cycle 1 OBs and showed less damage to the virion and the OB (paper in preparation). Furthermore, UV light can directly damage the DNA of the virus. A level of resistance to such UV damage can be conferred by DNA repair enzymes, as demonstrated by the increased virulence of UV-treated AcMNPV, following expression of an algal virus pyrimidine-dimer specific glycosylase [56]. In addition, Group II NPVs have conserved DNA photolyase genes, identified in NPVs isolated from *Chrysodeixis chalcites* and *Trichoplusia ni* larvae [57,58,59]. The DNA repair function of the Chrysodeixis chalcites NPV photolyase gene was confirmed by its expression in photolyase deficient *Escherichia coli*, conferring photo-reactivating ability [57,59]. Baculovirus cyclobutene pyrimidine dimer (CPD) photolyase (*phr*) genes may have been obtained from an ancestral lepidopteran insect host, as homologues have been identified in the lepidopteran insects *C. chalcites*, *Spondoptera exigua*, and *T. ni* [60].

The main objective of this work was to select a UV-resistant isolate of CrleGV-SA that would provide longer lasting virulence in the field as a biopesticide to improve its use within a management program for *T. leucotreta* [53]. We have demonstrated in laboratory conditions that the cycle 5 isolate from our selection process has significantly lower LC_50_ values compared to samples from earlier cycles, following UV exposure, and that with each re-exposure cycle the LC_50_ values moved closer to the value of the unexposed control. Sequence differences between the original CrleGV-SA and the UV-resistant isolate have been identified by NGS and REN analysis, indicating that the new isolate contains mutations either generated by the UV exposure process or selected from a pre-existing minor sub-population of the parent CrleGV-SA isolate. We consider the second scenario to be the most likely, considering the relatively low occurrence of SNPs. Consequently, what was a pre-existing minor sub-population in the wild-type isolate was selected to become the majority population through the selective pressure of repeated UV exposure. As this resistance to UV radiation is a competitive advantage for the virus, the existence of a non-competitive trade-off must be considered to explain why the genotype for resistance is not naturally dominant. No such trade-off could be identified, such as decreased production of virus. However, the explanation could lie in the lack of selection pressure in a natural environment. The virus population that will be the most persistent in nature, and thus most likely to be passed on and become the dominant population, is the population that will be protected from UV radiation, rather than the population that is exposed to UV radiation, even with a degree of UV tolerance. Furthermore, pest feeding takes place very often in these shaded and protected areas of the plant, which further increases the probability that virus in these protected areas will be predominantly ingested, propagated, and passed on, becoming the dominant virus population despite its lack of UV tolerance.

Future work is required to test the new isolate in the field and to confirm the genetic homogeneity and stability of the isolate population, particularly when passaged in vivo, as would be required for bulking up of virus for commercial field application. Such a study is currently underway.

## Figures and Tables

**Table 1 viruses-14-00028-t001:** LC_50_ values derived from bioassay data of virus samples exposed to UV in five exposure cycles.

Selection Cycle	UV Exposure Time (h)	LC_50_ (OB mL^−1^)	95% Fiducial Limits	
			Lower	Upper
**Cycle 1**	**0**	2.29 × 10^4^	3.37 × 10^−2^	4.73 × 10^4^
	1	3.96 × 10^4^	1.00 × 10^0^	6.00 × 10^5^
	3	8.97 × 10^5^	2.75 × 10^4^	4.60 × 10^7^
	8	4.73 × 10^7^	1.48 × 10^7^	1.39 × 10^8^
	24	2.89 × 10^8^	8.44 × 10^7^	1.17 × 10^9^
	72	2.11 × 10^9^	3.64 × 10^8^	1.69 × 10^11^
**Cycle 2**	0	2.57 × 10^4^	5.54 × 10^5^	9.24 × 10^−3^
	1	2.83 × 10^5^	4.55 × 10^2^	2.42 × 10^6^
	3	8.67 × 10^6^	1.01 × 10^6^	3.81 × 10^7^
	8	4.93 × 10^7^	1.27 × 10^7^	1.89 × 10^8^
	24	1.91 × 10^8^	5.58 × 10^7^	8.72 × 10^8^
	72	1.59 × 10^9^	3.66 × 10^8^	5.27 × 10^10^
**Cycle 3**	0	2.06 × 10^4^	3.69 × 10^−2^	4.04 × 10^5^
	1	1.30 × 10^5^	4.42 × 10^−1^	1.73 × 10^6^
	3	1.18 × 10^6^	1.62 × 10^4^	7.35 × 10^6^
	8	4.26 × 10^5^	1.99 × 10^5^	2.30 × 10^7^
	24	1.15 × 10^7^	2.71 × 10^6^	3.75 × 10^7^
	72	8.18 × 10^6^	1.97 × 10^5^	6.64 × 10^7^
**Cycle 4**	0	2.08 × 10^4^	1.89 × 10^−6^	8.32 × 10^5^
	1	1.47 × 10^6^	1.35 × 10^4^	9.60 × 10^6^
	3	5.36 × 10^5^	2.18 × 10^1^	6.90 × 10^6^
	8	1.22 × 10^7^	8.16 × 10^5^	5.75 × 10^7^
	24	4.12 × 10^5^	7.42 × 10^2^	3.59 × 10^6^
	72	6.12 × 10^6^	7.37 × 10^5^	2.18 × 10^7^
**Cycle 5**	0	2.87 × 10^4^	1.70 × 10^−5^	9.37 × 10^5^
	1	4.64 × 10^4^	2.07 × 10^−3^	1.27 × 10^6^
	3	1.93 × 10^5^	1.80 × 10^1^	2.57 × 10^6^
	8	6.38 × 10^6^	4.14 × 10^4^	4.61 × 10^7^
	24	2.16 × 10^5^	2.19 × 10^−1^	3.96 × 10^6^
	72	1.73 × 10^6^	1.60 × 10^4^	1.12 × 10^7^

**Table 2 viruses-14-00028-t002:** Relative potency comparisons (LC_50_) over a time course of UV exposure between CrleGV-SA samples from cycles 1 to 5 in surface concentration response bioassays against neonate *T. leucotreta* larvae using cycle 1 samples as the reference.

	Cycle 1 (Reference)	Non-Irradiated Control	Cycle 2	Cycle 3	Cycle 4	Cycle 5
1 h	1	0.198	7.846	6.599	29.494	2.683
3 h	1	0.086	12.298	2.205	2.345	0.536
8 h	1	0.001	1.213	0.134	0.318	0.293
24 h	1	0.000	0.699	0.027	0.002	0.004
72 h	1	0.000	1.262	0.004	0.001	0.001

**Table 3 viruses-14-00028-t003:** SNPs identified in both the CrleGV-SA C1 and CrleGV-SA C5 genomes.

Nucleotide Position	Change	Codon Change	Polymorphism Type	Amino Acid Change	Protein Effect	Protein
434	G -> A	TGT -> TAT	SNP (transition)	C -> Y	Substitution	Granulin
94,086	G -> A	GAA -> AAA	SNP (transition)	E -> K	Substitution	ORF-109
38,194	T -> C	ATT -> ACT	SNP (transition)	I -> T	Substitution	PIF-2
104,574	T -> C	ATG -> ACG	SNP (transition)	M -> T	Start Codon Loss	ORF-120
79,840	T -> G	TTG -> GTG	SNP (transversion)	L -> V	Substitution	ORF-103
45,853	C -> T	GTG -> ATG	SNP (transition)	V -> M	Substitution	39K protein
36,843	A -> T	TTT -> ATT	SNP (transversion)	F -> I	Substitution	Metallo-proteinase

**Table 4 viruses-14-00028-t004:** Seven new SNPs identified in the CrleGV-SA C5 genome.

Nucleotide Position	Change	Codon Change	Polymorphism Type	Amino Acid Change	Protein Effect	Protein
78,522	G -> A	TCT -> TTT	SNP (transition)	S -> F	Substitution	VP-91
59,752	A -> C	CAT -> CAG	SNP (transversion)	H -> Q	Substitution	DNA binding protein
59,752	A -> C		SNP (transversion)		Extension	ORF-73
59,779	T -> C	AGT -> GGT	SNP (transition)	S -> G	Substitution	ORF-73
59,709	TT -> CC	TTA, ATT -> TTG, GTT	Substitution	LI -> LV	Substitution	DNA binding protein
13,168	C -> T	GCT -> GTT	SNP (transition)	A -> V	Substitution	ORF-19
59,734	A -> T	GAT -> GAA	SNP (transversion)	D -> E	Substitution	DNA binding protein

## Data Availability

Publicly available datasets were analyzed in this study. This data can be found here: Mwanza, P. Development of a UV-tolerant strain of the South African isolate of Cryptophlebia leucotreta granulovirus for use as an enhanced biopesticide for *Thaumatotibia leucotreta* control on citrus. Ph.D. Thesis, Nelson Mandela University, 2020.

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
