# Peer review of "Selection for and Analysis of UV-Resistant Cryptophlebia Leucotreta Granulovirus-SA as a Biopesticide for Thaumatotibia leucotreta"

_viruses, 2021, doi:10.3390/v14010028_

Round 1

Reviewer 1 Report

See file

Author Response

We note the positive comments from reviewer 1 stating that: the paper addresses an important topic of improving UV-resistance of a widely used baculovirus. UV-induced damage is a major issue affecting the application of baculoviruses. The manuscript is in general well written with solid experiments and analyses.

In response to the specific comments:

  1. Where the viruses that are not damaged by UV light are called ‘survivors’ (line 17, 110, 129, 172) has been changed to “viruses that retained infectivity” or a similar phrase.
  2. Line 91-94 has been revised for clarification as follows:

“In field studies following spraying of CrleGV-SA biopesticide in citrus orchards, Mwanza [15] recorded 21 days after spraying, a significantly lower virus LD50 in samples from the southern (shady) side of citrus trees than on the northern (sunny) side of the trees. One week later (28 days after spraying), virulent virus was detected in samples from the northern side of the trees, but the activity was too low to quantify. In comparison, virus samples from the southern side of trees gave a clear bioassay dose response.”

  1. Line 139, the term pure has been removed and the source of virus clarified as follows in line 144:

“Fifth instar T. leucotreta larvae were inoculated with the known isolate of CrleGV-SA used in a formulation of the biopesticide, at the appropriate LC90 concentration [38].”

  1. The origin of the virus isolate from the commercial biopesticides is clarified in line 107 as follows:

“Several isolates of CrleGV have been identified. A natural isolate of CrleGV was first identified from the Ivory Coast (CrleGV-IC) [30], another isolate from Cape Verde (-CV) [31] and at least seven genetically distinct CrleGV-SA isolates from South Africa (-SA), one of which is used in the formulation of the commercial biopesticide produced in South Africa [32].”

  1. Line 178 has been revised as follows:

“After each exposure to UV light, the resultant virus sample was fed to fifth instars and the infective virus amplified within the larvae. Individual larvae were reared on artificial diet with a surface inoculation of 50 µl UV exposed CrleGV-SA in 25 well bioassay plates incubated at 30 °C. Infected larvae, dead and almost dead, were collected over a 14 day period and stored at -20 °C. Virus was extracted from the larvae and purified as described in section 2.1 and stored at 4 °C until the next exposure cycle or for analysis. The process of UV exposure and subsequent propagation of infective virus in fifth instars constituted one exposure cycle. A total of five exposure cycles were carried out.”

  1. Line 185: Larvae do not finish the diet but are indeed infected with the intended concentration of the virus. Please note that this is a tried and tested method with this and other insect-virus systems (e.g. MOORE, SD, HENDRY, DA & RICHARDS, GI. 2011. Virulence of a South African isolate of the Cryptophlebia leucotreta granulovirus (CrleGV-SA) to Thaumatotibia leucotreta neonate larvae. BioControl 56:341-352.). Furthermore, the technique is recognised to determine lethal concentrations (i.e. concentration applied), rather than lethal doses (i.e. dose consumed). We have now referenced this publication in the manuscript, line 189:

“Surface dose-response bioassays were conducted on first instars in 25 well bioassay plates, according to a standard protocol [38].”

  1. Section 2.6: what was the depth of sequencing? Also 3.2: why were there many more reads for the C5 isolate compared to the C1 isolate?

The sequencing coverage has been added to the results section along with Quality scores (Q40, Q30 and Q20) (see revised sections 2.6 and 3.2). We are not certain why C5 produced so many more reads compared to C1. The values in the manuscript have been corrected, with C5 producing more than 26 million reads compared to 416 thousand for C1. The values in the manuscript were the amount following error correction and normalisation. A commercial company assisted in the sequencing, with each sample possibly multiplexed differently, resulting in an unequal partition of sequencing capacity.

  1. Line 239: were the dead larvae in the control also showing symptoms of virus infection? Are these assumed to arise from covert infections?

Clarified in line 253:

“The dead larvae in these control bioassays did not exhibit symptoms of viral infection and were not considered to be covert infections.”

  1. Section 3.2: The assembled C5 isolate is more than 2000 bp longer than the C1 and published SA isolate (both same length) – it should be clarified where this difference comes from! In the discussion, line 410-411, it is said that no deletions or insertions were found, but that seems not correct given the difference in length of the isolates. Deletions or insertions may greatly impact certain genes in the genome.

This was a mistake introduced during our data review, with this number of 113,730 bp representing the length of a separate contig, rather than the length of the final consensus sequence for C5. The length of C5 has been double checked (along with other values) and these values corrected in the text as required. All other values reported have been checked and are correct (see revised section 3.2)

  1. Why was a restriction analysis included? Section 2.7 and 3.3

The restriction analysis has been removed.

  1. It is not addressed whether the UV irradiation changes the LT values of the virus isolates. This would be an important characteristic to test, since the time to death is highly relevant in biocontrol applications. This aspect should at least be addressed in the discussion.

Addressed in the discussion, line 421:

“The LT values for the UV-exposed viruses from cycle 1 and 5 were not determined in these bioassays, however this will be addressed in future experiments, since the time to death of the larvae is an important factor in biocontrol applications.”

  1. It is mentioned that UV damages the virion and the OB structure (line 428 and 429). One important aspect of UV is that the DNA of the viruses is damaged as well, it would be informative to add that. Some baculoviruses encode photolyase enzymes to repair such UV damage (e.g. Biernat et al., Insect Mol. Biol. 2011; van Oers et al, DNA Repair, 2008).

Note: we had described the effect of UV on DNA in the introduction in lines 58-62:

“UV irradiation damages baculoviruses by cross linking adjacent pyrimidine residues in the DNA, resulting in inhibition of DNA replication, mutation and blocking transcription factors [16,17]. Direct DNA damage also causes deletions, strand breakage and labile sites in the DNA [18]. In addition, UV-generated reactive oxygen species such as peroxides, single oxygen, or hydroxyl radicals inactivate the OBs [19,20].”

The following is added to the discussion, line 462-472:

Furthermore, UV light can directly damage the DNA of the virus. A level of resistance to such UV damage can be conferred by DNA repair enzymes, as demonstrated by the increased virulence of UV-treated AcMNPV, following expression of an algal virus pyrimidine-dimer specific glycosylase [56]. In addition, Group II NPVs have conserved DNA photolyase genes, indentified in NPVs isolated from Chrysodeixis chalcites and Trichoplusia ni larvae [57,58,59]. The DNA repair function of the Chrysodeixis chalcites NPV photolyase gene was confirmed by its expression in photolyase deficient Escherichia coli, conferring photo-reactivating ability [57,59]. Baculovirus cyclobutene pyrimidine dimer (CPD) photolyase (phr) genes may have been obtained from an ancestral lepidopteran insect host, since homologues have been identified in the lepidopteran insects C. chalcites, Spondoptera exigua and T. ni [60].

Minor points have been addressed and corrected.

Reviewer 2 Report

The manuscript reported the increase of tolerance to UV of baculovirus CrleGV-SA. Through repeated cycle of UV irradiation and virus propagation, the authors observed a decreased reduction in virulence after UV treatment. They further analyzed the sequences of virus genome in attempt to identify the potential genetic mechanism. The work provided some interesting clues to understand the variation in UV resistance in baculovirus.

Major conerns:

  1. Fig.1 could be omitted as it was based on the data inTable 1.
  2. The results of genome sequencing were not fully and clearly presented. Apart from single nucleotide mutations, were there any indels in the UV-treated genome?
  3. The virus used for sequencing was not genetically clonal, which meant that genetic variations were highly likely within the virus population. This might make the interpretation of the sequencing data complicated. Did the authors observe any polymorphism in the genome?  They should passage the virus in the host for several rounds (without further UV treatment) and confirm that the passaged virus was still tolerant to UV, before carrying out the sequencing. 
  4. Fig 2 was directly obtained from the sequencing data, which contained no new information. The author should carry out a real restriction analysis, and compare the result with the result of silico analysis of Fig 2, to help verify the sequencing result.
  5. It would be interesting for the authors to include some morphological comparison of the occlusion body of the original and UV-tolerant viruses, since they mentioned in the introduction that morphological changes might be associated with UV tolerance .
  6. The writing need much improvement to make the paper clear and concise. The introduction part should not contain too many of the experimental designs and results. 

Author Response

We note the positive comments from reviewer 2 regarding the work providing interesting clues to understand the variation in UV resistance in baculovirus.

In response to specific comments:

  1. Fig.1 could be omitted as it was based on the data in Table 1. Fig. 1 is removed.
  2. The results of genome sequencing were not fully and clearly presented. Apart from single nucleotide mutations, were there any indels in the UV-treated genome?

Additional details regarding genome sequencing have been included in the Materials and Methods section (section 2.6). Furthermore, additional results have been added for the sequencing of CrleGV-SA C1 and C5 including mean coverage with variation and Quality scores (Q40, Q30, and Q20). An error with the length of the C5 genome has also been corrected, with the resulting consensus sequence 111,334 bp in length, not 113,730 bp in length. No Indels were identified (section 3.2).

  1. The virus used for sequencing was not genetically clonal, which meant that genetic variations were highly likely within the virus population. This might make the interpretation of the sequencing data complicated. Did the authors observe any polymorphism in the genome? They should passage the virus in the host for several rounds (without further UV treatment) and confirm that the passaged virus was still tolerant to UV, before carrying out the sequencing.

We did not observe polymorphism in the genome and have inserted a comment to this effect in the results section (line 323).

We have added the following comment in the discussion regarding passage of the virus in the host for several rounds (without further UV treatment) and confirmation that the passaged virus was still tolerant to UV (line 456-460).

“The virus used for sequencing was not genetically clonal, but no polymorphism was observed in the genomes of cycle 1 or cycle 5 virus. Several rounds of passage of the cycle 5 UV-tolerant virus in the larval host, without further UV treatment, would confirm that it is a single isolate population. Re-testing for virulence after a further UV exposure would then confirm that the passaged virus was still tolerant to UV.”

  1. The sections on in silico restriction mapping and the figure 2 have been removed, in response to the advice by Reviewer 1.
  2. We refer to our TEM data, however this forms a separate paper in preparation, but we now make reference to a published extended abstract Mwanza, P.; Dealtry, G.B.; Lee, M.; Moore, S. Transmission electron microscopy analysis of changes to Cryptophlebia leucotreta granulovirus following ultraviolet exposure. Microscopy Society of Southern Africa Proceedings, 2017, 47, 78.
  3. The writing need much improvement to make the paper clear and concise. The introduction part should not contain too many of the experimental designs and results.

We have edited the introduction and have deleted all reference to the experimental design and results.

Reviewer 3 Report

This is a concise account of the selection of a baculovirus insecticide with enhanced resistance to UV light under laboratory conditions.  After five cycles of exposure to UV there was a significant reduction in LC50.  The bioassay data presented are sound .  The only thing lacking is more information on whether the mutations in the virus DNA after UV exposure account for these differences.  However, to map the significance of these mutations will be a major task and cannot be expected for this paper.

There is also a doubt as to whether or not the improvement in virus performance can be maintained in the field.  I look forward to the results from future field trials.

Author Response

We note the reviewer’s positive comments regarding the selection of a baculovirus insecticide with enhanced resistance to UV light under laboratory conditions, with bioassay data to indicate the significant improvement in LC50 and their acceptance of the manuscript.

The reviewer comments that more information is needed to determine whether the mutations in the virus DNA after UV exposure account for the resistance to UV and the improved LC50. As indicated in the manuscript discussion, this is the basis of further research, along with determination of the performance of the virus in the field.